# Atomic Layer Deposition of Chlorine Containing Titanium–Zinc Oxide Nanofilms Using the Supercycle Approach

Denis Nazarov [1,2,*], Lada Kozlova [1], Aida Rudakova [1], Elena Zemtsova [1], Natalia Yudintceva [3], Elizaveta Ovcharenko [3], Alexandra Koroleva [1], Igor Kasatkin [1], Ludmila Kraeva [4], Elizaveta Rogacheva [4] and Maxim Maximov [2]

1   Saint Petersburg State University, Universitetskaya Nab., 7/9, Saint Petersburg 199034, Russia; lada.kozlova.20@mail.ru (L.K.); aida.rudakova@spbu.ru (A.R.); ezimtsova@yandex.ru (E.Z.); koroleva.alexandra.22@gmail.com (A.K.); igor.kasatkin@spbu.ru (I.K.)
2   Peter the Great Saint Petersburg Polytechnic University, Polytechnicheskaya, 29, Saint Petersburg 195221, Russia; maximspbstu@mail.ru
3   Institute of Cytology of the Russian Academy of Sciences (RAS), Tikhoretsky Ave., 4, Saint Petersburg 194064, Russia; yudintceva@mail.ru (N.Y.); yelyzaveta.ovcha@mail.ru (E.O.)
4   Saint-Petersburg Pasteur Institute of Epidemiology and Microbiology, 14 Mira Street, Saint Petersburg 197101, Russia; lykraeva@yandex.ru (L.K.); elizvla@yandex.ru (E.R.)
*   Correspondence: dennazar1@yandex.ru; Tel.: +7-812-428-4033

**Abstract:** Atomic layer deposition (ALD) is a useful tool for producing ultrathin films and coatings of complex composition with high thickness control for a wide range of applications. In this study, the growth of zinc–titanium oxide nanofilms was investigated. Diethyl zinc, titanium tetrachloride, and water were used as precursors. The supercycle approach was used, and wide $ZnO/TiO_2$ (ZTO) ALD cycles were prepared: 5/1, 3/1, 2/1, 1/1, 1/2, 1/3, 1/5, 1/10, 1/20. Spectral ellipsometry, X-ray reflectometry, X-ray diffraction, scanning electron microscopy, SEM-EDX, and contact angle measurements were used to characterize the thickness, morphology, and composition of the films. The results show that the thicknesses of the coatings differ considerably from those calculated using the rule of mixtures. At high $ZnO/TiO_2$ ratios, the thickness is much lower than expected and with increasing titanium oxide content the thickness increases significantly. The surface of the ZTO samples contains a significant amount of chlorine in the form of zinc chloride and an excessive amount of titanium. The evaluation of the antibacterial properties showed significant activity of the ZTO–1/1 sample against antibiotic-resistant strains and no negative effect on the morphology and adhesion of human mesenchymal stem cells. These results suggest that by tuning the surface composition of ALD-derived ZTO samples, it may be possible to obtain a multi-functional material for use in medical applications.

**Keywords:** atomic layer deposition; thin films; titanium oxide; zinc oxide; ternary oxides; zinc titanium oxide



## 1. Introduction

Atomic layer deposition (ALD) is a well-known technology for growing ultrathin films and depositing coatings with high purity, conformality uniformity, and thickness control on the surfaces of various substrates. Due to these features, ALD has found widespread applications in a variety of scientific and industrial fields, such as microelectronics [1–3], catalysis [4], photocatalysis [5], solar energy [6,7], sensors [8,9], lithium-ion batteries [10–12], biomedicine [13,14], and its application areas are constantly expanding.

ALD is based on the chemical reactions of surface species with gaseous reagents. Because the reactions are sequential, the gaseous reagents do not mix and the number of reactive surface species is always limited, the chemical reactions are self-limiting, and a

fixed number of molecules are always deposited. In this way, excellent thickness control of the coatings can be achieved. However, one of the key advantages of ALD is the ability to control the composition of the films. Because the chemical reactions in ALD are cyclic, it is possible to change the nature of the chemical reactions and produce doped, multilayer films as well as films with complex compositions [15,16].

There are many approaches to producing multi-component compounds by ALD. These include the simultaneous injection of two non-interacting precursors, and the synthesis of multilayer films followed by heat treatment to allow interdiffusion and reaction of the components in the film [15]. Another alternative approach for the ALD of multi-component films is the use of multicomponent precursors, also known as heteronuclear precursors. Such precursors contain multiple metal centers or a metal with a non-metal that is intended to be incorporated into the film [15]. However, the most common and best way to control the composition and structure of the film is the supercycle approach. In this approach, the normal ALD cycles of sequential precursor and co-reactant pulses for each constituent process are combined into a cycle of cycles called a supercycle [15]. In this case, the composition of the film can be varied by changing the cycle ratio of simple compounds (normal ALD cycles) in a supercycle.

Complex, mixed, and doped zinc and titanium oxides (hereafter referred to as ZTO) obtained by the ALD process have broad prospects for use as transparent conductive oxides (TCO) [17–19], gas sensors [20] surface modification of cathode materials for lithium-ion batteries [21], electrode coatings for photocatalytic water splitting [22–24], a as well as protective coatings with corrosion resistance [25] and excellent tribological properties [26]. As titanium oxide is a widely used biocompatible material and zinc oxide is an oxide with bactericidal properties, ZTO may find applications in medical implants [27].

Pure titanium and zinc oxides are among the most successful and popular oxide systems synthesized by ALD. However, studies on the synthesis and characterization of their complex oxides or nanolaminates are scarce and the results presented by different authors are rather contradictory. To date, the results of such oxide systems have been reported in just over ten papers (Table 1). All of them use diethyl zinc (DEZ) and water as zinc oxide precursors. For the synthesis of the second component, titanium oxide, both titanium tetrachloride ($TiCl_4$) and metalorganic precursors—tetrakis(dimethylamido)titanium(IV) (TDMAT) and titanium(IV) isopropoxide (TTIP)—have been used.

**Table 1.** Summarized literature data about ZTO film growth and composition.

| Nominal ZnO/TiO$_2$ Ratio | Temperature/ Substrate | Thickness/Growth Rate | Composition | Ref. |
|---|---|---|---|---|
| DEZ-TiCl$_4$-H$_2$O precursors | | | | |
| 200/2, 200/4, 200/6, 200/8 | 170 °C/glass and Si (100) | 112–117 nm/not discussed | Close to theoretical | [28] |
| 1/200, 1/100, 1/50, 1/25 | 100 °C/Si, polycarbonate membrane | 28 nm/slight increase in growth rate by Zn doping | Zn content increase on the surface Presence of Cl | [29] |
| 13/1, 14/1, 15/1 | 120, 160, 200, 240 °C/quartz and silicon | 53–82 nm depending on deposition temperature (theoretical thickness—100 nm) | The real Ti concentration (4%) is more than the nominal one (2%) | [30] |
| DEZ-TTIP-H$_2$O precursors | | | | |
| 5/1, 10/1, 15/1, 20/1, 25/1, 30/1 | 200 °C/single-crystalline Si (100), soda-lime glass | 200 nm/the growth rates were slightly below the expected values | The linear correlation between pulses ration and Ti content up to 9.1% of pulse TTIP. A slight deviation at 16.7% pulse TTIP. | [17] |

**Table 1.** *Cont.*

| Nominal ZnO/TiO$_2$ Ratio | Temperature/ Substrate | Thickness/Growth Rate | Composition | Ref. |
|---|---|---|---|---|
| 1/1, 5/1, 10/1, 20/1, 25/1 | 200 °C/Corning glass | 50 nm/- | Nonlinear correlation Ti pulses and Ti content Ti doping is homogeneous | [31] |
| 1/1, 2/1, 5/1, 10/1, 20/1 | 200 °C/borosilicate glass, n-doped Si (100) | 9–148 nm/close to theoretical at low doping ratio and lower at high doping ratio (samples—ZTO 1/1, 2/1) | Order of the precursors' pulse affects the doping mechanism and film composition | [19] |
| 1/2 2/5 1/3 | 200 °C/p-type (100) Si | 160–185 nm/GPC of TiO$_2$ deposited on ZnO-terminated surface is faster than that of TiO$_2$, the GPC of ZnO on TiO$_2$-terminated surface is slower | Zn/Ti cycle ratio effect on the film composition is weak | [32] |
| 20/1, 10/1, 5/1, 2/1, 1/1 | 200 °C/ Si, thermally grown SiO$_2$, quartz | 80–106 nm/thicknesses were thinner than expected for samples with cycle ratio of ZnO/TiO$_2$ less than 10 | - | [33] |
| From 99/1 to 4/1 (9 types) | 200 °C/Thermally oxidized SiO$_2$ (100 nm) on Si | 50–58 nm/growth rates were higher than the estimated films exhibited a layer-by-layer structure | Ti concentration is more than expected. Adsorption of TTIP on the ZnO surface was enhanced relative to the TiO$_2$ | [34] |
| | | DEZ-TDMAT-H$_2$O precursors | | |
| 1/1, 1/2, 1/3, 1/4 | 90 °C/p-type (100) Si | 8.3–43.8 nm/thickness is less than expected and the difference increases with loop cycles number | Experimental and theoretical composition are close | [35, 36] |
| 1/1 | 200 °C/Si(100) wafers and glass substrates | A nucleation delay for ZnO deposited on the Si and TiO$_2$. TiO$_2$ growth on ZnO is greater than that of pure TiO$_2$ | Lower concentration of Ti$^{3+}$ comparing to pure TiO$_2$ and multilayered TiO$_2$/ZnO | [37] |
| 2/1 | 200 °C/Si(100) | 170, 1100 nm | Zn/Ti = 1/1 (Ti concentration is more than expected) The coating is oxygen deficient | [38] |

However, synthesis using chlorides has only been described in three papers (Table 1). Su et al. investigated the effect of zinc doping on the crystal structure and composition of the titanium oxide [29]. Torrisi et al. and Felizco et al. studied the doping of zinc oxide with titanium oxide at concentrations of a few percent [28,30]. There are no studies available to date that show results of the ALD with similar zinc/titanium oxide ratios in the supercycle.

In this study, we have investigated the growth characteristics of a ZTO system using supercycles in a wide range of ZnO and TiO$_2$ cycle ratios and using TiCl$_4$ as the titanium precursor. TiCl$_4$ is readily available, provides a relatively high growth rate, and can be successfully used over a wide temperature range, which is most important in the ALD of complex oxide systems. Due to the high reactivity and the absence of carbon in the precursor composition, the resulting titanium oxide is of high purity with very low chlorine content [39,40]. However, our results showed that ZTO contains much more chlorine than TiO$_2$. Despite this, a preliminary evaluation of the antibacterial properties and the effect of the coatings on the primary cytological response of human mesenchymal stem cells showed the potential promise of such coatings for obtaining medical materials.

## 2. Materials and Methods

### 2.1. Atomic Layer Deposition of TiO$_2$, ZnO, and ZTO Nanocoatings

TiO$_2$, ZnO, and ZTO thin films were fabricated by atomic ALD using a Nanoserf setup (Nanoengineering Ltd., Saint Petersburg, Russia). TiCl$_4$ (99%, Merck, NJ, USA) and Zn(C$_2$H$_5$)$_2$ (DEZ, 99.999% Elmos, Moscow, Russia) were used as Ti and Zn precursors, respectively, and deionized water as oxygen source. Coatings were deposited onto

monocrystalline silicon wafers (100) (diameters—40 mm and 100 mm) and cover glasses (18 × 18 mm). Nitrogen (99.9999%) was used as a carrier gas to transport precursors into the reaction chamber and as purging gas. $TiO_2$ and $ZnO$ films were deposited at 200 °C. During deposition of $TiO_2$ coating, pulse times were 100 ms for $TiCl_4$ and deionized water, purge time between precursor pulses was 10 s. For $ZnO$ deposition, pulse times were 50 ms for DEZ and 100 ms for deionized water, purge time between precursor pulses was 10 s. $TiCl_4$ or DEZ pulse, purge, $H_2O$ pulse, and second purge make up one cycle ALD. The number of cycles was 222 and 720 for $ZnO$ and $TiO_2$ deposition, respectively.

The complex zinc–titanium oxide (ZTO) nanofilm depositions were performed with use of the supercycle approach which included cycles of $TiO_2$ and $ZnO$. The ALD recipe can be described as follows:

$$\text{ZTO-n/k} = m \times (n \times ZnO + k \times TiO_2), \tag{1}$$

where m is the number of supercycles while n and k are the ratios of $ZnO$ and $TiO_2$ cycles in one supercycle. The former varied in a range of values from 31 to 170 (Table 2), whereas for the latter, values of 5/1, 3/1, 2/1, 1/1, 1/2, 1/3, 1/5, 1/10, and 1/20 were selected. The pulse times of precursors for the $TiCl_4/H_2O$ and $DEZ/H_2O$ processes were the same as for pure $ZnO$ and $TiO_2$ ALD. The number of supercycles (m) needed to deposit the coating with total thicknesses of 10 and 40 nm was calculated by the equation:

$$\text{Thickness} = m \times (n \times GPC_{ZnO} + k \times GPC_{TiO2}), \tag{2}$$

where GPC is the growth per cycle of the pure oxides, and n and k are the number of $ZnO$ and $TiO_2$ cycles in one supercycle. The calculated number of cycles and supercycles for 40 nm films are presented in Table 2.

**Table 2.** The rule of mixtures-based calculation of number of ALD cycles and supercycles for 40 nm film deposition.

| Composition of Supercycle (ZnO/TiO$_2$ Pulse Ratio) | Number of Supercycles | Number of Cycles | Estimated Thickness *, nm |
|:---:|:---:|:---:|:---:|
| 1/0 | 222 | 222 | 40.0 |
| 5/1 | 42 | 252 | 40.1 |
| 3/1 | 67 | 268 | 39.9 |
| 2/1 | 96 | 288 | 39.8 |
| 1/1 | 170 | 340 | 40.0 |
| 1/2 | 138 | 414 | 40.0 |
| 1/3 | 116 | 464 | 40.0 |
| 1/5 | 88 | 525 | 40.0 |
| 1/10 | 55 | 605 | 40.2 |
| 1/20 | 31 | 651 | 39.7 |
| 0/1 | 720 | 720 | 39.6 |

* The thickness values expected from the linear combination of GPC $TiO_2$ and GPC $ZnO$.

## 2.2. Samples Characterization

The thickness of $TiO_2$, $ZnO$, and ZTO coatings was determined by spectroscopic ellipsometry using Ellips-1891 SAG instrument (CNT, Novosibirsk, Russia). X-ray reflectometry (D8 DISCOVER, Bruker, Billerica, MA, USA) was used to investigate the thickness, density, and roughness of coatings. XRR analysis was conducted in the range of angles from 0.3° to 5° with an increment of 0.01 using symmetric scattering geometry. The results were processed by the simplex method using LEPTOS 7.7.

X-ray diffraction (XRD) was explored to characterize the crystallinity and phase structures of the $TiO_2$, ZnO, and ZTO coatings. XRD analysis was carried out using Bruker D8 DISCOVER (Bruker, Billerica, MA, USA) high-resolution diffractometer with monochromatic Cu Kα radiation for 2θ from 20° to 70° with a scan step 0.05°.

Scanning electron microscopy (SEM) was performed to observe the surface morphology of samples. The SEM imagery was obtained with Zeiss Merlin (Carl Zeiss, Oberkochen, Germany) using two regimes: SE (secondary electrons) and In-lens. Accelerating voltage was 20 kV. In-lens and EDS regimes at 10 kV were used for elemental analysis. The surface chemical composition of samples was evaluated by X-ray photoelectron spectroscopy (XPS). The measurements were obtained in a Thermo Fisher Scientific Escalab 250Xi photoelectron spectrometer (Thermo Fisher, Waltham, MA, USA) using monochromatic Al Kα source (1486.6 eV). Argon ion sputtering was used to remove the contamination from the surface of samples. The binding energy scale was compensated using C1s peak at 284.8 eV.

The contact angles (CA) were obtained by the sessile drop method using Theta Lite optical tensiometer (Biolin Scientific, Gothenburg, Sweden). Droplets of water were dispensed onto the surface of the samples using a microsyringe. The volume of water droplets was no more than 2.2 μL. During 10 s, the contact angles were calculated with a time step of 0.04 s. The value of the contact angles corresponding to a time of 10 s was taken as the final result. The mean value of CA for the same sample was determined by averaging at least 12 repeated measurements.

### 2.3. Antibacterial Activity

The antibacterial properties of the obtained samples were studied in accordance with ISO 22196:2011 (measurement of antibacterial activity on the surface of plastics and other non-porous materials) with the Gram-positive and Gram-negative multidrug-resistant bacteria: *Staphylococcus aureus*, *Acinetobacter baumannii*, *Pseudomonas aeruginosa*. Before the studies, the samples were not subjected to additional processing, including sterilization. The samples were placed in a 5 mL tube. Suspensions of microorganisms with a concentration of $10^6$ and $10^7$ CFU/mL were prepared. After that, 1 mL of control solutions were inoculated on nutrient medium. The samples were incubated in a thermostat at 37 °C for 24 h, then stored in a refrigerator at +2–4 °C until the results were obtained. The modified structures were placed in a suspension of microorganisms with a volume of 700 μL. The samples were covered with a sterile piece of parafilm (40 × 40 mm) and incubated in a thermostat with a closed lid at 37 °C for 24 h. Next, 100 μL of the test samples were moved to separate sterile tubes with 9 mL of 0.9% NaCl for 1 h. Then, 1 mL of the saline solution was inoculated on nutrient medium. The samples were incubated in a thermostat at 37 °C for 24 h. Finally, the number of colonies was calculated.

### 2.4. FetMSC Cells Adhesion

Human fetal mesenchymal stem cells derived from bone marrow (FetMSCs) were obtained from the shared research facility "Vertebrate cell culture collection", supported by the Ministry of Science and Higher Education of the Russian Federation (Agreement No. 075-15-2021-683). FetMSCs were previously seeded on coverslips with different coatings ZnO, ZTO–1/1, ZTO–1/20, and $TiO_2$) and left to cultivate at 37 °C overnight (24 h). Then the samples were fixed with 4% paraformaldehyde in PBS and permeabilized with 0.1% Triton X-100 in phosphate buffer solution (PBS). To visualize the actin cytoskeleton, cells were stained with Rhodamine phalloidin (Invitrogen, Waltham, MA, USA) for 20 min. Nuclei were stained and mounted into the fluorescent mounting medium supplemented with 4′,6-diamidino-2-phenylindole (DAPI) (Abcam, Cambridge, UK). Cells were analyzed using a microscope (Olympus FV3000) with an Olympus IX83 confocal system (Olympus Corp., Tokyo, Japan). Nuclei and actin cytoskeleton were detected with diode lasers 405 nm and 561 nm, respectively.

## 3. Results

### 3.1. Atomic Layer Deposition

Based on the linear dependence of thickness on the number of ALD cycles (Figure S1a) and growth per cycle of pure $TiO_2$ and ZnO, the number of supercycles for the ZTO films of 10 and 40 nm thickness was calculated according to the rule of mixtures (Table 2). These films were deposited on the silicon surface. The film thicknesses differ significantly from the calculated ones, as shown by the ellipsometric results (Figure 1). Non-monotonic dependencies were obtained. In the case of a small proportion of titanium oxide cycles in the total supercycle, the film thicknesses are significantly lower than those calculated. Starting from sample ZTO–3/1, the thicknesses increase and are practically comparable to the calculated thicknesses with an equal ratio of $TiO_2$ and ZnO cycles (sample ZTO–1/1). With a higher proportion of $TiO_2$ cycles, the thicknesses significantly exceed the theoretical ones and reach a maximum for sample ZTO–1/10.

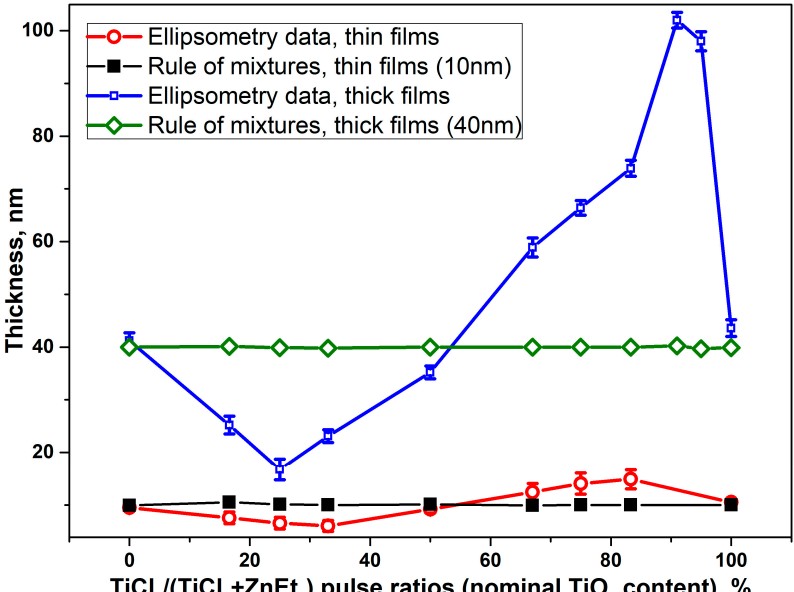

**Figure 1.** The difference between the theoretical (calculated) and real thickness of coatings depending on the composition of supercycles.

Based on the GPC data obtained for different ZTO supercycle compositions and near-linear dependence of thickness on the number of ALD cycles (Figure S1b), films with a thickness of about 40 nm were deposited on the silicon surface. The thickness values obtained by ellipsometry and X-ray reflectometry are presented in Table 3. According to the ellipsometry data, the thicknesses are close to 40 nm and range from 38.2 to 43.6 nm. The thickness values of ZnO and ZTO–5/1 samples obtained from the XRR data are 1–3 nm higher than the ellipsometry values, but for samples with high titanium content, the thicknesses were 2–5 nm lower than the ellipsometry values. Unfortunately, an attempt to determine the exact thickness of the films using cross-sectional SEM images was unsuccessful due to the films being too thin, insufficient interface contrast, and edge effects. The samples with high zinc content (ZnO, ZTO–5/1, and ZTO–3/1) are characterized by the highest roughness (>2 nm). For the other samples, the roughness is minimal and does not exceed 1 nm. The density values of the pure oxide layers are close to those of the bulk materials (ZnO—5.65 $g/cm^3$, anatase $TiO_2$—4.05 $g/cm^3$). Sample ZTO–5/1 has the lowest density and the highest roughness, which may be due to the peculiarity of the morphology and will be further confirmed by the SEM data.

**Table 3.** XRR and spectral ellipsometry data of 40 nm films deposited on the silicon *.

| Sample | Thickness (Ellipsometry), nm | Thickness (XRR), nm | Roughness, nm | Density, g/cm$^3$ |
|---|---|---|---|---|
| ZnO | 42 ± 0.5 | 43.2 | 2.43 | 5.88 |
| ZTO–5/1 | 43.1 ± 0.4 | 45.2 | 6.28 | 3.76 |
| ZTO–3/1 | 43.6 ± 0.6 | 37.9 | 2.03 | 5.15 |
| ZTO–1/1 | 38.2 ± 0.3 | 34.7 | 0.26 | 3.96 |
| ZTO–1/3 | 41.5 ± 0.4 | 39.1 | 0.89 | 4.1 |
| ZTO–1/5 | 40.4 ± 0.5 | 36.5 | 1.08 | 3.88 |
| ZTO–1/20 | 40.2 ± 0.3 | 37.0 | 0.95 | 3.87 |
| TiO$_2$ | 43.7 ± 0.7 | 37.7 | 0.67 | 4.11 |

* The silicon substrate (50 × 15 mm).

### 3.2. Structure and Morphology of the Films

The X-ray diffraction patterns for as-deposited and annealed at 350 °C samples are shown in Figure 2. The as-deposited ZnO sample showed the presence of reflections in the region of 31.7, 34.4, and 36.2° corresponding to the 100, 002, and 101 planes of the hexagonal wurtzite (ZnO, zincite) crystal structure. The relatively high intensity of the 002 orientation indicates that the coating prefers to grow with the c-axis perpendicular to the support surface [30]. Additionally, the two cristobalite (SiO$_2$) peaks 101 and 202 appear for ZTO–5/1 and ZTO–3/1 samples. The presence of cristobalite is caused by the presence of defects in the silicon substrate. For the remaining as-prepared samples, only peaks corresponding to silicon were detected. After thermal treatment at 350 °C, the 102 and 110 wurtzite planes appear for ZnO, but the 002 orientation remains dominant. The samples of the ZTO series calcined at 350 °C were amorphous. However, the annealed TiO$_2$ sample shows an anatase structure with clearly visible peaks of 101, 103, 004, 101, 200, and 105.

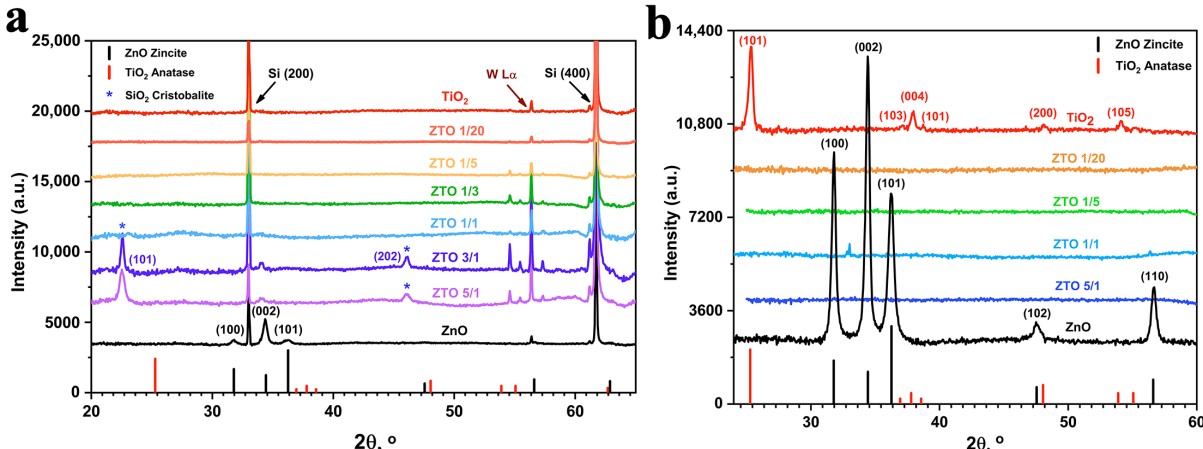

**Figure 2.** XRD patterns of the as-prepared (**a**) and annealed at 350 °C (**b**) samples.

The morphology of the films was investigated by scanning electron microscopy (Figure 3). High-resolution SEM images showed that the ZnO surface consists of densely packed uniform elongated nanograins, which is typical for ALD-grown ZnO films deposited on silicon surface [30,41]. ZTO–5/1 contains larger grains of size 100–300 nm which are not densely packed (Figures 3, 4 and S2). The grains of sample ZTO–3/1 become even larger and their density lower. For the remaining samples, there are no large grains, but round particles ranging in size from a few tens of nanometers to several hundred. For samples with lower zinc content (ZTO–1/1, ZTO–1/3, ZTO–1/5) large particles of several hundred nanometres are typical. Samples ZTO–1/10, ZTO–1/20 and especially pure TiO$_2$

are characterized by the presence of small particles (less than 100 nm). Similar particles have been previously observed in $TiO_2$ obtained by ALD on the surface of silicon and titanium, representing $TiO_2$ with an anatase structure [40,42]. However, due to their low abundance, anatase was not detected by XRD. According to local EDX analysis of the composition of the coarse particles, they contain significant amounts of zinc and chlorine. It can therefore be assumed that they are partly or wholly composed of zinc chloride. However, zinc and chlorine are also detected in particle-free areas, but in smaller amounts. It is noteworthy that the composition of the different particles varies considerably for the ZTO–1/1 sample and the range of composition variation is shown in Figure 4d.

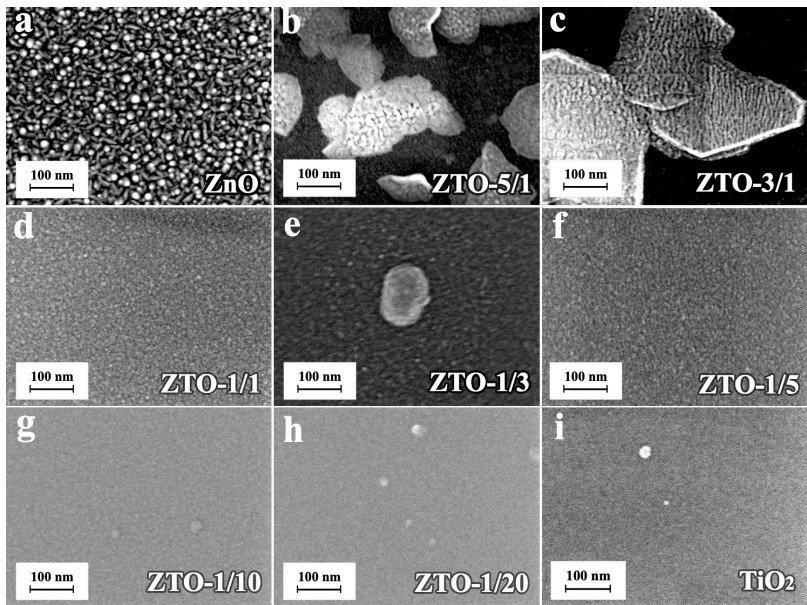

**Figure 3.** Plan-view SEM images of ZnO, $TiO_2$, and ZTO samples at magnification of 300,000. (**a**) ZnO, (**b**) ZTO–5/1 (**c**) ZTO–3/1, (**d**) ZTO–1/1 (**e**) ZTO–1/3 (**f**) ZTO–1/5 (**g**) ZTO–1/10 (**h**) ZTO–1/20 (**i**) $TiO_2$.

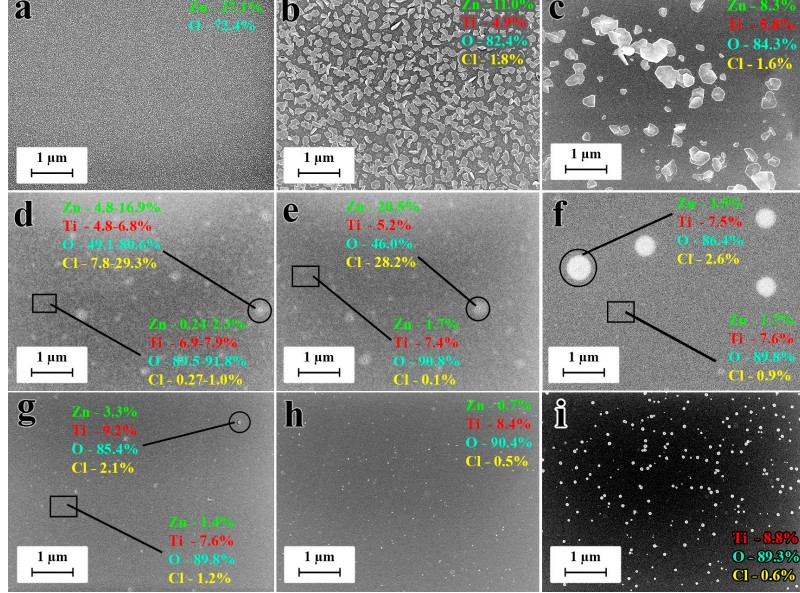

**Figure 4.** Plan-view SEM images and results of local EDX analysis of ZnO, $TiO_2$, and ZTO samples at magnification 30,000. (**a**) ZnO, (**b**) ZTO–5/1 (**c**) ZTO–3/1, (**d**) ZTO–1/1 (**e**) ZTO–1/3 (**f**) ZTO–1/5 (**g**) ZTO–1/10 (**h**) ZTO–1/20 (**i**) $TiO_2$. The circles indicate the areas of the surface composition study by the EDX containing particles, and the rectangles are the areas without particles.

### 3.3. Surface Composition and Wettability of the Nanocoatings

Quantitative analysis of the surface composition was carried out using XPS data (Figure 5). The content of zinc and titanium naturally increases as the proportion of ZnO and TiO$_2$ cycles in the supercycle increases. However, the change in the concentration of these elements is not monotonic. For example, for samples ZTO–2/1, ZTO–1/1, and ZTO–1/2, the zinc content is almost the same (about 14%–14.6%) and the titanium content varies insignificantly (13%–15%). As the TiO$_2$ cycle fraction increases, the oxygen content also increases. Apart from pure oxygen, all samples also contain significant amounts of chlorine. The highest chlorine content was found for ZTO–5/1. For the other samples with high ZnO cycles (ZTO—3/1 to ZTO–1/2), the chlorine content is almost constant (4.5%–5.1%). The lowest amount of chlorine (~2%) was found for the sample with the lowest proportion of ZnO in the supercycle (ZTO–1/20) The presence of carbon is caused by the adsorption of carbonaceous compounds during storage of the samples in an air atmosphere (so-called adventitious carbon). This assumption is confirmed by the fact that the lowest carbon content was found in the last two samples (ZTO–1/20, ZTO–1/10), which were stored in the air for the shortest time.

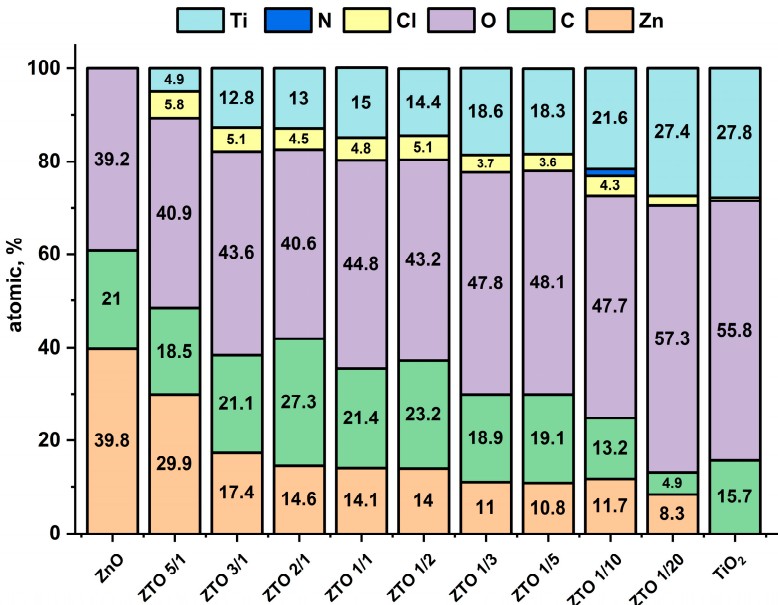

**Figure 5.** Results of quantitative analysis of surface composition measured by XPS.

The C1s high-resolution XPS data for all samples show complex multi-component peaks (Figure 6a). Deconvolution of these spectra can be performed using three or four components with maxima at 284.8, 286.3, 287.8, and 288.8 eV. The example of the three-component deconvolution of ZTO–1/1 is shown in Figure S3a. The most intense peak for all samples at 284.8 eV corresponds to the aliphatic hydrocarbons or other organic molecules with C–C and C–H bonds. The less intense (286.3 eV) corresponds to C–OH and/or C–O groups. The peaks at around 288.8 and 287.8 eV correspond to the carboxyl (O–C=O) and aldehyde (C=O) groups.

The O1s spectra (Figure 6b) of the samples show three overlapping peaks. The example of the three-component deconvolution of ZTO–1/1 is shown in Figure S3b. The most intense peaks at 529.8 and 529.9 eV belong to the oxygen of ZnO and TiO$_2$, respectively. However, for sample ZTO–5/1 the peak at higher energy is more intense. It generally corresponds to hydroxides, surface hydroxyl groups, defective or organic oxygen, and oxygen in non-stoichiometric titania (TiOx) [43,44]. The third peak with a maximum of 532.7 eV corresponds to oxygen from organic surface impurities (C–O, C=O). The presence of zinc chloride can be confirmed by Cl1s peaks consisting of two Zn-Cl components (Figures 6c and S3e).

The Ti2p spectra (Figure 6d) for all samples show intense peaks of Ti2p3/2 and Ti2p1/2 at 458.7–458.9 and 464.4–464.6 eV. The doublet splitting for Ti2p3/2 and Ti2p1/2 is 5.7 eV (Figure S3c), corresponding to $TiO_2$ [45]. The Zn2p spectra (Figure 6e) for all samples show intense peaks of Zn2p3/2 and Zn2p1/2 with a doublet splitting of 23.1 eV (Figure S3d), corresponding to ZnO. However, the positional peaks of only the ZnO sample correspond to ZnO maxima at 1021 and 1044.1 eV. For all other samples, there is a shift towards higher energy. This shift may be due to the presence of $ZnCl_2$ [46]. Surprisingly, the deconvolution did not show the presence of other components besides ZnO and $TiO_2$ (Figure S3c,d).

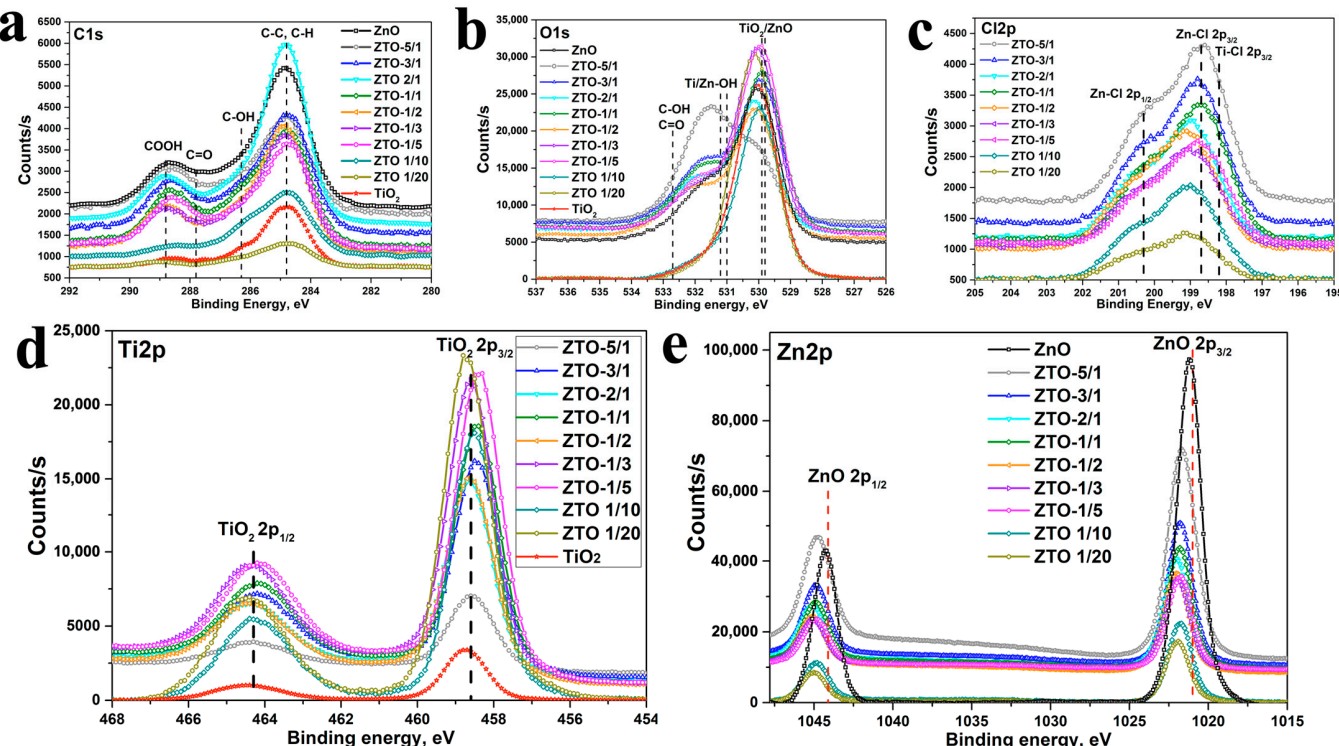

**Figure 6.** XPS spectra of all studied coatings (**a**) C1s, (**b**) O1s, (**c**) Cl2p, (**d**) Ti2p, and (**e**) Zn2p.

In order to assess the hydrophilicity of the samples studied, the water contact angles were measured (Figure 7). All samples are slightly hydrophilic (contact angle less than 90°), but pure $TiO_2$ and ZnO oxides are more hydrophobic than ZTO samples. No statistical difference was observed between the different ZTO samples.

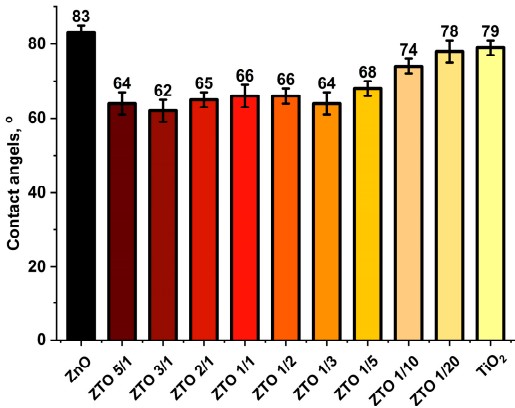

**Figure 7.** Results of water contact angle measurements. Data are presented as mean ± C.I. from at least 12 independent series of experiments ($p < 0.05$) in different areas of the samples.

The bactericidal properties of the samples were analyzed using a strain of one of the most persistent hospital pathogens: *Staphylococcus aureus*, *Acinetobacter baumannii,* and *Pseudomonas aeruginosa*. No bacterial growth was detected on the ZnO and ZTO–1/1 samples for *S. aureus* and *A. baumannii* and relatively little growth for *P. aeruginosa* (Table 4). A slight advantage was observed for ZnO compared to ZTO–1/1 for *P. aeruginosa*. The TiO$_2$ sample showed antibacterial activity only against *S. aureus*.

**Table 4.** Results of antibacterial study of samples as the number of colony forming units (CFU).

| Strain | ZnO | ZTO–1/1 | TiO$_2$ | Control |
|---|---|---|---|---|
| *S. aureus* | No growth | No growth | ~10 | $1 \times 10^7$ |
| *A. baumannii* | No growth | No growth | $1 \times 10^6$ | $1 \times 10^6$ |
| *P. aeruginosa* | ~100 | ~1000 | $1 \times 10^7$ | $1 \times 10^6$ |

FetMSC cells were used to study the effect of coating on cell adhesion and morphology. TiO$_2$, ZnO, ZTO–1/1, and ZTO–1/20 coatings with a thickness of 40 nm were deposited on the surface of the coverslips. After 24 h of cultivation, the samples were examined by fluorescence microscopy. On the surface of all samples except ZnO, the cells showed good adhesion and spreading (Figure 8). The cells on all these samples were characterized by the formation of a series of extensions called filopodia. The cells are "connected" to each other by these filopodia and show the typical morphology of MSCs [42]. The cells grown on the ZnO surface were different. They have no clear filopodia and are poorly spread.

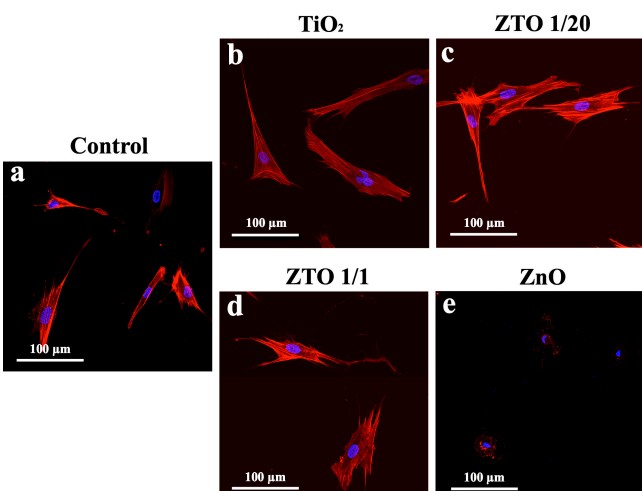

**Figure 8.** Immunofluorescent images of FetMSC cells cultivated on the cover glasses coated by ALD. FetMSc actin cytoskeleton (red), nuclei (blue). (**a**) control sample without coatings, (**b**) TiO$_2$, (**c**) ZTO–1/20, (**d**) ZTO–1/1, and (**e**) ZnO.

## 4. Discussion

The results showed that the growth rates of the ZTO layers differ significantly from those calculated theoretically by the rule of mixtures. Due to the different crystal structures and different ionic radii of Zn$^{2+}$ (0.074 nm) and Ti$^{4+}$ (0.068 nm), TiO$_2$ and ZnO are not able to dope each other with significant amounts of [29]. Moreover, binary oxides such as Zn$_2$TiO$_4$ and ZnTiO$_3$ are only formed after annealing at temperatures above 650 °C [32,47]. In [30], the presence of Ti-containing nanodots with an average diameter of 2 nm homogeneously distributed in the film was detected at a ratio of ZnO to TiO$_2$ of 14/1. In this context, the growth characteristics of the ZTO system should be considered mainly from the point of view of the growth of separate phases of TiO$_2$ and ZnO.

As the literature analysis shows, the influence of TiO$_2$ and ZnO growth on each other is complicated. Despite the number of papers, the results are different when using the same

reagents. For example, Refs. [32,32] show a significant decrease in the growth rate of ZnO doped with $TiO_2$, while [34] shows a slight stimulation. However, in these works, TTIP was used as a precursor. In our case, $TiCl_4$ was used and there may be a significant effect of reactions using HCl as one of the reagents. In the $TiO_2$ production cycle, reactions (3) and (4) take place where HCl is produced. As the process is carried out in an inert gas stream, most of the HCl is removed from the surface, but some can interact with ZnO to form $ZnCl_2$ (5), which partially evaporates under synthesis conditions (200 °C, 13 mm Hg) and the film thickness is much less than expected. Since zinc chloride is a low volatile reagent, some of it remains on the surface and we therefore observe significant chlorine content on the surface of the ZTO samples. The ZTO–1/1 sample, obtained at a higher temperature of 300 °C, contains considerably less chlorine on the surface, which may indirectly confirm the hypothesis of evaporation of the formed $ZnCl_2$ from the surface.

$$S\text{-OH (s)} + TiCl_4 \text{ (g)} = S\text{-O-}TiCl_{4-n} \text{ (s)} + nHCl \text{ (g)} \tag{3}$$

$$S\text{-O-}TiCl_{4-n} \text{ (s)} + {}_{4-n}H_2O \text{ (g)} = S\text{-O-Ti(OH)}_{4-n} \text{ (s)} + {}_{4-n}HCl \text{ (g)} \tag{4}$$

$$2HCl \text{ (g)} + ZnO \text{ (s)} = ZnCl_2 + H_2O \text{ (g)}, \tag{5}$$

where S—is the surface.

Similar growth inhibition and an elevated titanium content compared to zinc were observed in studies in which TTIP [17,19,32,33] and TDMAT [35–37] were used instead of chloride. Doyle et al. report about a nucleation delay for ZnO deposited directly on the $TiO_2$ [37]. One reason could be the lower number of –OH groups on the $TiO_2$ surface compared to ZnO [34]. Thus, the resulting titanium oxide is an inhibitor of zinc oxide growth. For the same reason, an increase in GPC is possible for ZTO samples with large amounts of titanium chloride pulses, where ZnO plays the role of a kind of catalyst with a higher number of hydroxyls. Increased growth of zinc-doped titanium oxide was observed in other works [29,32]. The influence of the number of reactive hydroxyl groups can explain the growth features at low and high ratios of the number of cycles of ZnO and $TiO_2$ in the supercycle. At low ratios, i.e., frequent ZnO and $TiO_2$ cycle changes, the interaction of zinc oxide with hydrogen chloride formed during the adsorption of titanium chloride plays a greater role. Partial removal of the growing zinc oxide in the form of chloride could also explain the deviation of the experimental Ti/Zn ratio from the theoretical one (Figure 9).

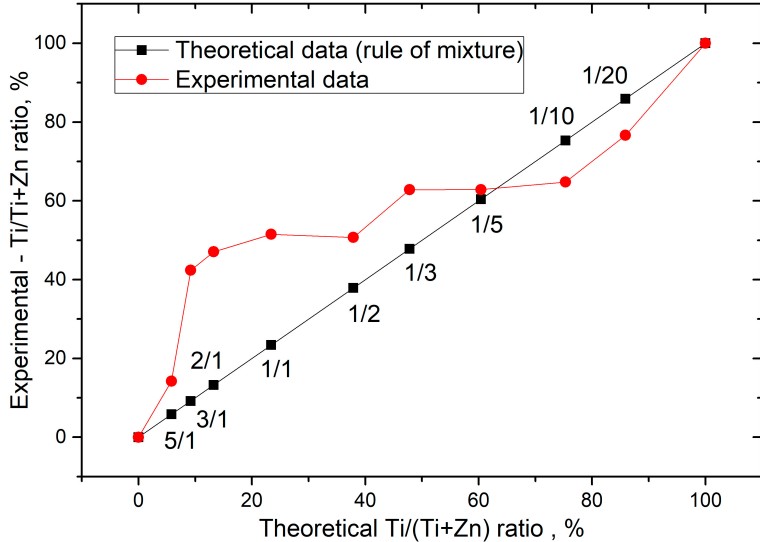

**Figure 9.** Comparison of the experimental and the theoretical composition of the surface calculated according to the rule of mixture.

Unfortunately, XRD did not provide any new information on the structure of the coatings. The crystalline phase was only determined for pure ZnO and annealed $TiO_2$. A number of studies have shown a gradual change in crystal structure during ALD doping of prepared ZnO, but at titanium concentrations above 10%, the films become completely amorphous [31]. For $TiO_2$ doped with ZnO ($ZnO/TiO_2 < 1/25$), crystallization should take place at temperatures above 300 °C, but all the ZTO studied in our work have a higher $ZnO/TiO_2$ ratio.

Two complementary methods were used to analyze the composition of the coatings. XPS provides information mainly about the surface layer (4–10 nm), whereas the depth of information of EDX is about 0.5–2 μm. In other words, XPS provides information about the surface of the samples, while EDX provides information about the bulk but local average composition (several μm in diameter). Unfortunately, the resolution of EDX is not sufficient to resolve the difference in the bulk composition of the dense submicron and nanometer particles in samples ZTO–5/1 and ZTO–3/1 but allows for resolving the difference in the bulk composition of the rounded particles on the surface of the other ZTO samples (Figure 4d–h). Additionally, based on the EDX and XPS data, it can be assumed that zinc chloride is predominantly concentrated in these particles or that these particles consist of zinc chloride. The surface which is from particles consists of titanium and zinc oxides. Chlorine and carbon are mainly found on the surface. After sputtering the surface layer with argon ions (0.5 keV, 60 s) of sample ZTO–1/1, the amount of chlorine and carbon is less than 2%.

The presence of fairly polar $ZnCl_2$ on the surface has obviously influenced the increased hydrophilicity of ZTO samples compared to pure oxides. However, surface hydrophilicity depends on many factors such as the band structure of the oxides, the amount and type of molecules adsorbed on the surface, the freshness of the samples [42], and for films tens of nanometers thick, the substrate has a significant influence [48]. Wetting angle variations of ALD ZTO samples have only been demonstrated only for one sample in the paper by Doyle et al. and the results are controversial [37].

A large number of studies have demonstrated the high biocompatibility of ALD titanium oxide [49–51] and the antibacterial properties of zinc oxide [52–54]. Antimicrobial activity and biocompatibility are usually difficult to combine, as antibacterial compounds in most cases exhibit cytological activity. However, fine-tuning and adjusting the composition of ZTO layers, which is easily achieved by the ALD technique, can solve this problem. In our study, we performed a preliminary evaluation of the antibacterial activity of ZTO–1/1 samples as well as the pure oxides ZnO and $TiO_2$. The results were quite as expected. ZnO showed the highest activity, but this sample had a negative effect on MSC cell adhesion and spreading. ZTO–1/1 showed slightly weaker antibacterial activity, but still proved to be biocompatible with regard to cell adhesion and spreading. It should be noted that even the presence of chlorine on the surface had no significant negative effect on the cytological response. Cell adhesion and spreading are the first and one of the most important steps in the biointegration of the material. Surface properties such as wettability, surface energy, topography, and surface composition have a major influence on cell response. In this respect, our proposed approach to tune the surface composition of the material could be promising, but further detailed studies using more biocompatibility assessment techniques, more different samples, and detailed statistical analysis are required to confirm this.

## 5. Conclusions

The results obtained from the studies showed that the thicknesses of the coatings differ significantly from those calculated using the rule of mixtures. The minimum growth rate was observed for ZTO–3/1 and the maximum for ZTO–1/10. These results suggest that a different growth mechanism takes place for ZTO compared to pure ZnO and $TiO_2$. The decrease in growth rate for the samples with relatively high ZnO cycles is caused by the reaction of the by-product of the $TiO_2$ cycle—HCl with ZnO. This reaction leads to the partial elimination of ZnO and contamination of the films with chlorine as $ZnCl_2$. A

reduction in the proportion of ZnO in the supercycle reduces the elimination of zinc oxide, while the catalytic effect of a small amount of zinc oxide leads to a significant increase in the growth of $TiO_2$. The data also suggest that the presence of zinc chloride increases surface hydrophilicity. The surface and bulk compositions of the coatings obtained also differ significantly from those expected from the rule of mixtures. However, the results show that the composition and morphology of the coatings can be adjusted over a wide range by changing the composition of the supercycle. Summarizing these features, the evaluation of the antibacterial properties and the nature of the adhesion of the mesenchymal stem cells, we can draw a conclusion about the possible wide prospects for the use of ZTO obtained by ALD in the medical field.

**Supplementary Materials:** The following supporting information can be downloaded at: https://www.mdpi.com/article/10.3390/coatings13050960/s1, Figure S1: Thickness of ZnO and $TiO_2$ (a) and ZTO (b) films as a function of the number of ALD cycles. Figure S2: Plan-view SEM images of ZnO, $TiO_2$, and ZTO samples at magnification of 10,000. (a) ZnO, (b) ZTO–5/1 (c) ZTO–3/1, (d) ZTO–1/1 (e) ZTO–1/3 (f) ZTO–1/5 (g) ZTO–1/10 (h) ZTO–1/20 (i) $TiO_2$. Figure S3: XPS spectra of ZTO–1/1 (a) C1s, (b) O1s, (c) Ti2p, (d) Zn2p, and (e) Cl2p.

**Author Contributions:** Conceptualization, D.N.; methodology, D.N., N.Y., A.K., E.Z. and L.K. (Ludmila Kraeva); validation, D.N. and M.M.; formal analysis, D.N.; investigation, D.N., E.O., A.K., A.R., I.K., L.K. (Lada Kozlova) and E.R.; resources, D.N., M.M., N.Y., E.Z. and L.K. (Ludmila Kraeva); data curation, D.N. and L.K. (Lada Kozlova); writing—original draft preparation, D.N., N.Y., E.R. and L.K. (Lada Kozlova); writing—review and editing, D.N., A.R., E.Z., I.K. and M.M.; visualization, D.N., E.O. and L.K. (Lada Kozlova); supervision, D.N.; project administration, D.N.; funding acquisition, D.N. All authors have read and agreed to the published version of the manuscript.

**Funding:** The research was conducted under the financial support of the Russian Science Foundation grant (project No. 22-73-00093), https://rscf.ru/project/22-73-00093/ (accessed on 19 May 2023). Wettability measurements were conducted by Aida Rudakova with financial support from Saint Petersburg State University (Pure ID 94030186).

**Institutional Review Board Statement:** Not applicable.

**Informed Consent Statement:** Not applicable.

**Data Availability Statement:** The main data had been provided in the article and supplementary material. Any other raw/processed data required to reproduce the findings of this study are available from the corresponding author upon request.

**Acknowledgments:** This research was conducted using the equipment of the resource centers of the Research Park of the St. Petersburg State University Innovative Technologies of Composite Nanomaterials, Center for Physical Methods of Surface Investigation, Nanotechnology Interdisciplinary Center, X-ray Diffraction Studies and Nanophotonics.

**Conflicts of Interest:** The authors declare no conflict of interest.

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
