# Peer review of "Atomic Layer Deposition of Chlorine Containing Titanium–Zinc Oxide Nanofilms Using the Supercycle Approach"

_coatings, doi:10.3390/coatings13050960_

Round 1
Reviewer 1 Report
Review Report
The manuscript “Atomic layer deposition of chlorine-containing titanium-zinc oxide nanofilms using the supercycle approach” reports the preparation of ZnO/TiO2 (ZTO) thin films with various ZnO/TiO2 ratios by atomic layer deposition (ALD) in a supercycle mode and the assessments of their antibiotic and biocompatible properties. The results have shown that by varying the cycle number of the precursor, ZTO thin films with varying ZnO/TiO2 ratios can be successfully obtained. These ZnO/TiO2 ratios are found to impact the deviation of the experimentally-measured thin film thickness from the theoretically calculated value. Also, the existence of chlorine on the surface of ZTO thin films endows samples with good antibacterial properties. With further adjustments of the amount of TiO2, the antibacterial and biocompatible can be acquired in the ZTO thin films such as the ZTO-1/1 sample. This work has well demonstrated that ALD is a flexible method to prepare complex oxides with required properties, which would be of interest to the readership of Coating. However, when it comes to the publication of the current manuscript, there are still some questions needed to be clarified. So, I would recommend its publication after the authors have refined these concerns.
1. The “Introduction” part may need to be reorganized. In the current version, the necessities of preparing ZTO thin films and employing chloride-containing precursors are not clear. In other words, the authors did not state why authors want to prepare ZTO thin films and why use precursors different from the previous.
The paragraph, “In some cases, due to chemical, steric or thermodynamic limitation…is almost impossible.” (Lines 63-68) seems a little bit irrelevant.
2. As stated in Lines 360-367, titanium oxide is an inhibitor of zinc oxide growth. But this may not be used as the ground to explain the GPC increase for the Ti-rich ZTO samples. How to link the inhibitor effect of the titanium oxide to the catalytic effect of the zinc oxides?
Author Response
The manuscript “Atomic layer deposition of chlorine-containing titanium-zinc oxide nanofilms using the supercycle approach” reports the preparation of ZnO/TiO2 (ZTO) thin films with various ZnO/TiO2 ratios by atomic layer deposition (ALD) in a supercycle mode and the assessments of their antibiotic and biocompatible properties. The results have shown that by varying the cycle number of the precursor, ZTO thin films with varying ZnO/TiO2 ratios can be successfully obtained. These ZnO/TiO2 ratios are found to impact the deviation of the experimentally-measured thin film thickness from the theoretically calculated value. Also, the existence of chlorine on the surface of ZTO thin films endows samples with good antibacterial properties. With further adjustments of the amount of TiO2, the antibacterial and biocompatible can be acquired in the ZTO thin films such as the ZTO-1/1 sample. This work has well demonstrated that ALD is a flexible method to prepare complex oxides with required properties, which would be of interest to the readership of Coating. However, when it comes to the publication of the current manuscript, there are still some questions needed to be clarified. So, I would recommend its publication after the authors have refined these concerns.
- The “Introduction” part may need to be reorganized. In the current version, the necessities of preparing ZTO thin films and employing chloride-containing precursors are not clear. In other words, the authors did not state why authors want to prepare ZTO thin films and why use precursors different from the previous.
In fact, the production of chloride-containing films was not our task. Our task was to study the peculiarities of the synthesis of the ZTO system by the ALD and to evaluate the possibility of medical applications. And the discovery of such a large amount of chlorine was unexpected. It is well known that ALD titanium oxide films obtained from titanium chloride contain almost no chlorine. In addition, the available work on the preparation of ZTO using titanium chloride does not investigate its purity. Only one study [29] briefly mentions the presence of chlorine in the films. According to the literature and our experience, titanium chloride is not inferior to the other successful precursors for titanium oxide. Moreover TiCl4 is readily available, provides a relatively high growth rate and can be successfully used over a wide temperature range, which is most important in the ALD of complex oxide systems.
In fact, it turned out that there was a lot of chlorine in the ZTO, and we spent a lot of time looking for an explanation - why this is happening. Nevertheless, we have obtained very interesting results. We hope that they will be very useful to those who want to achieve ALD ZTO thin films for a wide variety of applications. And our preliminary in vitro results show that the presence of chlorine on the surface is not a negative factor, but may even be positive in terms of antibacterial activity.
To help the reader understand our aims and how they relate to the results, we have added a small explanation at the end of the introduction:
In this study we have investigated the growth characteristics of a ZTO system using supercycles in a wide range of ZnO and TiO2 cycle ratios and using TiCl4 as the titanium precursor. TiCl4 is readily available, provides a relatively high growth rate and can be successfully used over a wide temperature range, which is most important in the ALD of complex oxide systems. Due to the high reactivity and the absence of carbon in the precursor composition, the resulting titanium oxide is of high purity with very low chlorine content [39,40]. However, our results showed that ZTO contain much more chlorine than TiO2. Despite this, a preliminary evaluation of the antibacterial properties and the effect of the coatings on the primary cytological response of human mesenchymal stem cells showed the potential promise of such coatings for obtaining medical materials.
We have also slightly rewritten the introduction to make it clearer to the reader our motivation for choosing the ZTO system and its possible applications (4th paragraph):
Complex, mixed and doped zinc and titanium oxides (hereafter referred to as ZTO) obtained by the ALD process have broad prospects for use as transparent conductive oxides (TCO) [17-19], gas sensors [20] surface modification of cathode materials for lithium-ion batteries [21], electrode coatings for photocatalytic water splitting [22-24], а as well as protective coatings with corrosion resistance [25] and excellent tribological properties [26]. As titanium oxide is a widely used biocompatible material and zinc oxide is an oxide with bactericidal properties, ZTO may find applications in medical implants [27].
2. The paragraph, “In some cases, due to chemical, steric or thermodynamic limitation…is almost impossible.” (Lines 63-68) seems a little bit irrelevant.
In this paragraph we wanted to show that the supercycle approach is theoretically very nice and effective, but in practice there are many difficulties and limitations. We agree that it is not entirely appropriate for an introduction and have removed it from the text.
3. As stated in Lines 360-367, titanium oxide is an inhibitor of zinc oxide growth. But this may not be used as the ground to explain the GPC increase for the Ti-rich ZTO samples. How to link the inhibitor effect of the titanium oxide to the catalytic effect of the zinc oxides?
In this part of the discussion we explain the changes in growth rate in terms of chemical groups (particularly hydroxyl groups) available for chemical interaction during ALD. And our conclusion is quite simple. The growth of both oxides occurs by reaction of hydroxyl groups with TiCl4 and ZnEt2. According to the literature, a much greater number of reactive OH groups are formed on the surface during zinc oxide growth. This is one of the reasons why the GPC for pure ZnO is much higher than for TiO2 (0.180 vs 0.055 nm). Therefore, the addition of ZnO as part of the supercycle leads to an increase in the growth of TiO2 due to a higher amount of hydroxyl. Conversely, the growing TiO2 has fewer reactive hydroxyl groups, resulting in less growth per ZnO cycle. All this determines the growth of films at small and large ZnO/TiO2 ratios. When the ratio of ZnO and TiO2 cycles in a supercycle is almost equal (they often alternate in a supercycle), the interaction process of zinc oxide with hydrogen chloride plays a major role. In this case, a sufficiently large amount of zinc oxide is produced and a sufficient amount of etchant - HCl - are released for the etching process to have a significant effect on the film thickness. We write about this in the next paragraph.
We have slightly edited the text and merged the paragraphs to make it clearer for the reader.
Reviewer 2 Report
This article investigates the growth rates and morphologies of complex zinc-titanium oxide nanofilms prepared by atomic layer deposition. The ZTO-1/1 complex film shows a significant activity against antibiotic resistant strains and no negative effect on the morphology and adhesion of human mesenchymal stem cells. Major revision is recommended. The detailed comments are listed below:
1. The motivation should be emphasized in the introduction part. Why Cl is needed in the composite coatings? What are the advantages of composite coatings in medical applications?
2. When the Ti content in the composite coating increases, the thickness of the coating will decrease. Whether this is caused by the co-reactant HCl?
3. The linear growth of ZnO and TiO2 with the ALD cycles should be presented. How about the growth rate of ZTO with the supercycles?
4. It is interesting that the ZTO films are amorphous after 350 C annealing. However, there are large grains on the surface. Please explain the reason. How high temperature the crystallization can be happen at?
5. The SEM and EDX results show the films are not uniform with large particles. The thickness tested by ellipsometry and XRR should be calibrated by cross-view of the films.
Author Response
This article investigates the growth rates and morphologies of complex zinc-titanium oxide nanofilms prepared by atomic layer deposition. The ZTO-1/1 complex film shows a significant activity against antibiotic resistant strains and no negative effect on the morphology and adhesion of human mesenchymal stem cells. Major revision is recommended. The detailed comments are listed below:
- The motivation should be emphasized in the introduction part. Why Cl is needed in the composite coatings? What are the advantages of composite coatings in medical applications?
Thank you for your comment. It is indeed a question that the attentive reader may have.
In fact, the production of chloride-containing films was not our task. Our task was to study the peculiarities of the synthesis of the ZTO system by the ALD method and to evaluate the possibility of medical applications. And the discovery of such a large amount of chlorine was unexpected. It is well known that ALD titanium oxide films obtained from titanium chloride contain almost no chlorine. According to the literature and our experience, titanium chloride is not inferior to the other successful precursors for titanium oxide. In addition, the available work on the preparation of ZTO using titanium chloride does not investigate its purity. Only one study [29] briefly mentions the presence of chlorine in the films.
In fact, it turned out that there was a lot of chlorine in ALD ZTO, and we spent a lot of time looking for an explanation - why this is happening. Nevertheless, we have obtained very interesting results. We hope that they will be very useful to those who want to achieve ALD ZTO thin films for a wide variety of applications. And our preliminary in vitro results show that the presence of chlorine on the surface is not a negative factor, but may even be positive in terms of antibacterial activity.
To help the reader understand our aims and how they relate to the results, we have added an explanation at the end of the introduction:
In this study we have investigated the growth characteristics of a ZTO system using supercycles in a wide range of ZnO and TiO2 cycle ratios and using TiCl4 as the titanium precursor. TiCl4 is readily available, provides a relatively high growth rate and can be successfully used over a wide temperature range, which is most important in the ALD of complex oxide systems. Due to the high reactivity and the absence of carbon in the precursor composition, the resulting titanium oxide is of high purity with very low chlorine content [39,40]. However, our results showed that ZTO contain much more chlorine than TiO2. Despite this, a preliminary evaluation of the antibacterial properties and the effect of the coatings on the primary cytological response of human mesenchymal stem cells showed the potential promise of such coatings for obtaining medical materials.
In terms of the benefits of composites, successful biomaterials must have a very wide range of functional properties. This can be achieved by using composites that combine different properties. In our case, we wanted to combine the antibacterial properties of zinc oxide (zinc ions) with the high bioactivity of titanium oxide, which is considered to be one of the most successful biocompatible oxides used in orthopaedic and dental implants. We have slightly revised the text of the Introduction in the fourth paragraph to make this clearer:
Complex, mixed and doped zinc and titanium oxides (hereafter referred to as ZTO) obtained by the ALD process have broad prospects for use as transparent conductive oxides (TCO) [17-19], gas sensors [20] surface modification of cathode materials for lithium-ion batteries [21], electrode coatings for photocatalytic water splitting [22-24], а as well as protective coatings with corrosion resistance [25] and excellent tribological properties [26]. As titanium oxide is a widely used biocompatible material and zinc oxide is an oxide with bactericidal properties, ZTO may find applications in medical implants [27].
- When the Ti content in the composite coating increases, the thickness of the coating will decrease. Whether this is caused by the co-reactant HCl?
The growth rate of the ZTO in our process is determined by many factors. It is true that the HCl etches the formed TiO2 film. This reduces the growth per cycle and thus the overall thickness. But etching by HCl always occurs, and the tendency you mention is only observed with a very high TiO2 fraction and correspondingly low ZnO. So we think that in this case there is a reduction in the catalysing effect of the ZnO and probably less zinc chloride is formed on the surface. To make this clearer to the reader, we have rewritten the paragraph before Figure 9:
A similar growth inhibition and an elevated titanium content compared to zinc was observed in studies in which TTIP [17,19,32,33] and TDMAT [35-37] was used instead of chloride. Doyle et al. reports about a nucleation delay for ZnO deposited directly on the TiO2 [37]. One reason could be the lower number of -OH groups on the TiO2 surface compared to ZnO [34]. Thus the resulting titanium oxide is an inhibitor of zinc oxide growth. For the same reason an increase in GPC is possible for ZTO samples with large amounts of titanium chloride pulses, where ZnO plays the role of a kind of catalyst with higher number hydroxyls. Increased growth of zinc doped titanium oxide was observed in other works [29,32]. The influence of the number of reactive hydroxyl groups can explain the growth features at low and high ratios of the number of cycles of ZnO and TiO2 in the supercycle. At low ratios, i.e. frequent ZnO and TiO2 cycle changes, the interaction of zinc oxide with hydrogen chloride formed during the adsorption of titanium chloride plays a greater role. Partial removal of the growing zinc oxide in the form of chloride could also explain the deviation of the experimental Ti/Zn ratio from the theoretical one (Figure 9).
- The linear growth of ZnO and TiO2 with the ALD cycles should be presented. How about the growth rate of ZTO with the supercycles?
We have provided experimental data on the linear growth of ZnO and TiO2 in a supplementary file (Figure S1a). We have not done a focused study on the linear growth of the ZTO system, but we have enough data for different compositions of ZTO to assess the linear or near linear growth of ZTO for films of a few tens of nanometres. We have also shown this data in Figure S1b).
- It is interesting that the ZTO films are amorphous after 350 C annealing. However, there are large grains on the surface. Please explain the reason. How high temperature the crystallization can be happen at?
Yes, it is a very interesting question. We repeated our measurements several times on two XRD instruments and indeed our large grain samples were always amorphous. This is very strange indeed. The analysis of literature data did not give us any information. Perhaps the reason has to do specifically with the peculiarity of the ALD process, i.e. the use of the supercycle approach. Grains were only detected in samples ZTO-5/1 and ZTO-3/1. In their synthesis, 5 and 3 cycles of ZnO are always followed by a cycle of TiO2. During these 5 and 3 cycles, less than 1 nm of zinc oxide can grow, and the subsequent chemisorption of titanium oxide affects the possibility of long-range order formation and the formation of the crystal layer. However, this does not affect grain formation. It should also be noted that, according to SEM, these grains are quite porous and may contain titanium oxide in addition to zinc oxide. TiO2 is converted to volatile TiCl4 by interaction with HCl. Its evaporation leads to further amorphization of these grains without changing their shape. This is only a weak hypothesis and we do not know how it could be confirmed or refuted by the available data or possibilities. Therefore, we do not consider it necessary to present it in the text of the manuscript.
It is also difficult for us to say at what temperature crystallisation should occur, as for nanofilms this process is highly dependent on thickness and composition. It is known from literature data that titanium doped ZnO (i.e. ZTO with high Zn/Ti ratio) should be crystalline without calcination. The synthesis temperature of 200C should be sufficient for crystallisation [30], but at titanium concentrations above 10% the films become completely amorphous [31]. Therefore, the crystallisation of our films should take place at higher temperatures. For TiO2 doped with ZnO (ZnO/TiO2 < 1/25), crystallisation should take place at temperatures above 300C, but all the ZTO studied in our work have a higher ZnO/TiO2 ratio. [29]. Therefore, we chose 350C for annealing. Perhaps we could have obtained a crystalline film for ZTO-1/20 or even ZTO-1/10 by raising the temperature to 400C, but this needs to be tested and will take time. It is safe to say that crystallisation would occur at temperatures above 650C, but in this case the formation of Zn2TiO4 and ZnTiO3 would occur.
We mention this in the first paragraph of the discussion section. A small explanation of crystallisation has also been added in the paragraph after Figure 9:
Unfortunately, XRD did not provide any new information on the structure of the coatings. The crystalline phase was only determined for pure ZnO and annealed TiO2. A number of studies have shown a gradual change in crystal structure during ALD doping of prepared ZnO, but at titanium concentrations above 10% the films become completely amorphous [34]. For TiO2 doped with ZnO (ZnO/TiO2 < 1/25), crystallisation should take place at temperatures above 300 °C, but all the ZTO studied in our work have a higher ZnO/TiO2 ratio.
- The SEM and EDX results show the films are not uniform with large particles. The thickness tested by ellipsometry and XRR should be calibrated by cross-view of the films.
Indeed, cross-sectional SEM images can be the most descriptive and accurate method for determining thickness of the films. However, this is true for relatively thick films, typically in excess of 100 nm. Due to the limited resolution of microscopes and the manifestation of edge effects (https://www.jeol.com/words/semterms/20121024.012800.php#gsc.tab=0), it is difficult to achieve measurement accuracy better than 2-3 nm. In addition, our films are semiconducting and the substrate is semiconducting, so the contrast between them is very weak. This also makes it difficult to accurately determine the interface and measure the thickness. In our case, the films were about 40 nm thick, and therefore it is not possible to achieve an accuracy of more than a few nanometers. Approximately such discrepancy we observed between ellipsometry and XRR.
We have attempted to measure the thickness of ZTO and ZnO films using SEM by pre-cutting a small area on the surface (see attached in .doc file SEM images) using focused ion beam (FIB). In this case we used both the technique with and without the deposition of a carbon layer on the surface to improve contrast. Unfortunately, accurate and reliable results could not be obtained.
As an alternative method - we could use TEM. But there are two limitations. Firstly, it is a much more complicated and expensive technique to measure a whole series of samples of different composition and thickness. Secondly, it would be very difficult to achieve very high accuracy because of the amorphous silicon oxide layer on the silicon surface. It would be difficult to find a clear boundary between it and the amorphous layers. As a confirmation, I can quote data from our recent article where we investigated titanium oxide thin films by TEM [43 - doi:10.3390/coatings12050668].
We have also made ellipsometry calibrations using cross-sectional SEM data on thick films of pure zinc and titanium oxides. Some of the cross-sectional data have been quoted in a number of our articles and those of our colleagues who obtained the films on the same equipment and under the same conditions. [https://doi.org/10.1016/j.jallcom.2021.159746, DOI: 10.1134/S1063783415090036, doi:10.3390/coatings12050668]. The differences in SEM, ellipsometry and XRR for ZnO do not exceed 2-3%. TiO2 is different. Ellipsometry gives overestimated results. XRR gives underestimated results. The true values are somewhere in the middle. And the deviations between the data increase with increasing thickness and can reach 15%. As our thicknesses are small, we cannot calibrate accurately for thick films, but based on the fact that ellipsometry gives 43 nm and reflectometry gives 37 nm, the true thickness should be around 40 nm. The largest deviation we have is for these samples.
As for the influence of particles, they are rather flat and occupy only a small fraction of the area in most samples and should not affect the ellipsometry results significantly. The only exception is sample ZTO-5/1, which has a large number of submicron planar particles on its surface. However, when calculating the thickness of this sample by ellipsometry we used a model with a variable porosity of about 15-25% and we believe that the calculated thicknesses are close to the real values.
A small additional explanation of the SEM measurements of thicknesses has been added to the text in the paragraph after figure 1.

Round 2
Reviewer 1 Report
The authors have well explained the concerns and revised the manuscript. So I would recommend its publication.
Reviewer 2 Report
The revised manuscript can be accepted.